# Usage of Morphological Mutations for Improvement of a Garden Pea (*Pisum sativum*): The Experience of Breeding in Russia

**Andrey Sinjushin [1]** , **Elena Semenova [2],\*** and **Margarita Vishnyakova [2]**

[1] Department of Genetics, Faculty of Biology, Lomonosov Moscow State University, Leninskie Gory 1-12, 119234 Moscow, Russia; asinjushin@mail.ru

[2] N.I. Vavilov Institute of Plant Genetic Resources (VIR), 190000 St. Petersburg, Russia; m.vishnyakova.vir@gmail.com

\* Correspondence: e.semenova@vir.nw.ru

**Abstract:** The improvement of pea as a crop over many decades has been employing the use of mutants. Several hundreds of different mutations are known in pea (*Pisum sativum* subsp. *sativum*), some of which are valuable for breeding. Breeding strategies may be diverse in different countries depending on different obstacles. In Russia, numerous spontaneous and induced mutations have been implemented in breeding. To our knowledge some of these, are not used in pea breeding beyond Russia. This review describes the use of mutations in pea breeding in Russia. The paper provides examples of cultivars created on the basis of mutations affecting the development of seeds (*def*), inflorescence (*det*, *deh*), compound leaves (*af*, *af uni*^tac), and symbiotic nitrogen fixation (various alleles of *Sym* and *Nod* loci). Novel mutations which are potentially promising for breeding are currently being investigated. Together with numerous cultivars of dry and fodder pea carrying commonly known mutations, new 'chameleon' and 'lupinoid' morphotypes, both double mutants, are under study. A cultivar Triumph which increases the effectiveness of interactions with beneficial soil microbes, was bred in Russia for the first time in the history of legume breeding.

**Keywords:** afila; determinate growth; fasciation; morphotypes; nitrogen fixation; nodulation

## 1. Introduction

The diversity of living things results from their outstanding ability to change, especially due of heritable variations, i.e., mutations. Without a special survey, many biological species seem uniform, although an ongoing progress in genomics (including whole-genome studies in natural populations) indicates a level of genetic diversity [1,2]. A huge amount of this variation, however, is associated with the non-coding fraction of the genome, such as repetitive sequences, and, as it does not affect the phenotype, often remains unseen.

When any species is domesticated, its variation becomes the subject of detailed analysis. A certain part of this polymorphism, either arising spontaneously or induced, may be utilized for the further improvement of this crop or domestic animal. As a result, many species which were domesticated long ago display an outstanding range of phenotypic variation, which can be exemplified in both animals (pigs, horses, pigeons) and plants (wheat, maize, tomatoes, ornamental plants, pea), the latter being in a scope of a given review. This variation can be (and actually is) much wider than observed among the wild relatives of these animals and crops, as a vector of artificial selection often contradicts the supposed direction of natural selection and preserves forms, which are inadaptive but valuable for the purposes of breeding (e.g., [3]).

The origin of genetics is deservingly associated with Gregor Johann Mendel, who reported the results of his pea (*Pisum sativum* L.) crossing experiments in 1865 [4]. The outline of these events is well known, but it should be emphasized that Mendel did not induce

mutations, nor did any of plant breeders or naturalists of that age. All these variations appeared spontaneously but remained in genotypes of certain cultivars, sometimes long before Mendel began his work. For example, unpigmented seed testa and the associated white color of petals (mutation *a*) was first recorded ca. 1300 [5]. Two of Mendel's variations, fasciation and pods lacking a lignified inner layer, were mentioned as early as 1597 by Gerard [6].

As a result, hundreds of morphological mutations are known in pea [7], many of which have been localized on a genetic map. Most of them are valuable material for the dissection of regulatory pathways of such aspects as floral monosymmetry, compound leaves or symbiotic nitrogen fixation. Many of them were (and actually are) conventional markers for linkage mapping [8–13]. However, although it is a model object for genetics and physiology, pea is still of importance as a valuable crop species, one of the most significant in temperate climatic zones. This means that novel (or, in contrast, well known) heritable morphological anomalies can be rated as a source of valuable features for further crop improvement [14].

More than 150 years of study in the field of pea genetics has resulted in a somewhat paradoxical, probably unique situation. Studies in pea as both a crop and model species has led to formation of rich germplasm collections. In these collections, hundreds of mutants and thousands of cultivars and local races from across the globe are being stored and are available for work [14].

The accessions bearing already identified mutations have an outstanding value. Some aspects of genetics and physiology, such as the regulation of compound leaf ontogeny, are very precisely investigated mostly because numerous mutants are known. However, the pea genome is both very large (ca. 3.9 Gb in available genome assembly) and complicated for bioinformatics with its high abundance of transposable elements (67.1% of the whole assembly) [15]. It is therefore of no surprise, that its whole genome sequence became available much later than those of *Medicago truncatula* Gaertn., *Cajanus cajan* (L.) Millsp. or *Lupinus angustifolius* L., i.e., leguminous species which not nearly as well characterized with respect to their genetics. The paradox (and tragedy) of *P. sativum* is the deep gap between its perfect degree of development as a model in genetics and its poorly characterized features in tearms of genomics.

In this context, the case of the molecular identification of gene *A* (with its recessive mutation causing the 'Mendelian white' flower color) is especially exemplary. Although this phenotype is known for more than 700 years and its genetics was studied by G. Mendel, the identification of causative gene required analysis of syntenic region of chromosome of *M. truncatula* (already characterized with whole-genome sequence by that moment) and search of similarity between candidate gene and its putative orthologue in *Arabidopsis thaliana* (L.) Heynh. (Brassicaceae) [5].

The molecular identification of certain morphological mutations is not a primary goal of breeding (or it has not been so until recently), as mutant alleles with a known pattern of inheritance can be introduced into genotypes of cultivars without knowing their sequence. For an annual plant such as pea, many morphological peculiarities become visible soon after germination, so there is no need to know the exact molecular nature of mutation. However, in recent decades it has become evident, that information on the gene sequence may be very helpful in marker-assisted selection (MAS). Looking further forward, it should be noted that it is difficult to find another leguminous crop for which the mutation pool is used in breeding as actively as in peas.

The area of cultivation for peas is very wide, preferentially in temperate latitudes. Thousands of cultivars are known worldwide and are being bred continuously. During 2019–2021 only, the collection of the Vavilov Institute of Plant Genetic Resources (VIR, Russia) was supplemented by 87 new cultivars, and approximately half of them were of Russian origin. In different countries, strategies of breeding may be diverse depending on soils, climatic conditions, the level of development of breeding technologies, national cuisine, and available initial material, etc. The given review aims to characterize the Russian experience of pea breeding, focusing mostly on use of morphological mutations. As many

papers describing trends of breeding in the former USSR and Russia (Russian Federation) are available only in Russian, this review may be of interest for readers.

As there is no commonly accepted terminology regarding different types of cultivars, we will henceforward apply the following terms. 'Dry' peas are those harvested for fully mature dry (usually smooth) seeds; 'fodder' (or forage) cultivars are grown for feeding animals, while seeds of 'vegetable' cultivars are harvested immature green to be either consumed immediately or stored frozen or canned. Although taxonomic relations between different peas have been debated for a long time, we follow the system reviewed and summarized by Maxted and Ambrose [16]. A recent genome-wide investigation points to the possibility of two genetically independent domestication events [17]. However, for the sake of uniformity and in the absence of other evidences, we classify most Russian and foreign white-flowered dry and vegetable cultivars as the 'garden' variety *P. sativum* subsp. *sativum* var. *sativum*. Fodder cultivars, if they have pigmented flowers and seeds, can be identified as the 'field' variety, *P. sativum* subsp. *sativum* var. *arvense*. This dichotomy is based mostly on the presence of pigmentation, i.e., a monogenic habit, and seems quite confusing, as two cultivars, although formally belonging to the same subspecies and variety, may be very morphologically diverse regarding their seeds, leaves, and stems etc.

## 2. Seed and Pod

The improvement of plants through breeding is primarily targeted at increasing their productivity, together with (and often assisted by) decreasing losses during harvesting. One of features of wild-growing plants, which has been repeatedly reduced during the domestication of different species, is the existence of one or another mode of seed dispersal. In legumes, this is usually associated with pod shattering (dehiscence), which is considered a key marker distinguishing wild from cultivated forms of pea [18–20]. Moreover, the ability of mature seeds to shed (Figure 1A) has been characteristic feature of even cultivated peas for centuries, thus causing significant losses. The mutation preventing abscission of funiculus from hilum in the course of seed maturation was first discovered at Priekule breeding station, Latvia [21]. Seeds did not shed even from fully mature dehiscent pods (Figure 1B). As Vasikh Khangildin and Willy Khangildin [22] found out in later studies, this feature was characteristic for homozygotes for recessive mutation *def* (*development funiculus*). On cellular level, no abscission layer was formed on the boundary between a funiculus and seed hilum [23].

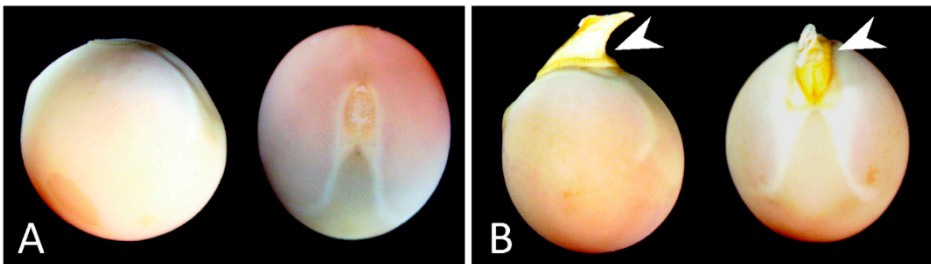

**Figure 1.** Pea seeds of abscising (**A**, cv. Topaz) and non-abscising (**B**, cv. Batrak) phenotypes. A persistent funiculus is indicated with an arrowhead.

Since the late 1960s, several cultivars featuring this mutation, such as Neosypayushchiysya 1, Pershotsvit, Truzhenik, and several others, have been bred at the Lugansk experimental station by the breeder Anatoly Shevchenko. The latter cultivar was included in the list of recommended cultivars in 1984 and is still cultivated in Russia. A series of cultivars lacking seed abscission, including Shirkhan, Chishminskii 75, and Chishminskii80, was produced in the 1980s at the Chishmy experimental station (Bashkortostan). Since then, this non-abscising seed habit has become very popular in Russian cultivars. As stated in [24], almost half of all contemporary Russian cultivars of pea have non-abscising (*def*) seeds, although there is still no agreement about the influence of this trait on actual seed losses

during harvesting. Hybrids and recombinants which are homozygous at *def* produce fewer seeds per pod than *Def* plants [24]. Some research indicates, that the intact non-abscising funiculus may delay water imbibition by seeds and hence affect their germination rate and cooking time [25]. Beyond Russia and the former USSR, mutation *def* seems not to be popular in pea breeding, probably because the persistent funiculus somewhat deteriorates the appearance of mature seeds (Figure 1B).

The inheritance of the best-known mutation causing a wrinkled seed shape was studied by G. Mendel, and this trait was the first Mendelian character identified on a molecular level. This mutation affects the gene encoding starch-branching enzyme, *Rugosus* (*R*). As a result of insufficient enzymatic activity, the synthesis of starch is diminished, so sucrose is accumulated in developing embryos. As a result seeds become sweeter, and more water is lost during their maturation, thus causing the wrinkling of their surface. This is a factor which defines the production of smooth versus wrinkled seeds. As seed shape is conditioned by carbohydrate content (more amylose in wrinkled seeds, reviewed in [26]), the genotype at *R* locus is a key feature distinguishing between dry (*R*) and vegetable cultivars of pea (*r*). To date, several mutant alleles are known in *R* locus [6]. In Russia, areas where vegetable peas are sown are expanding (21,572 ha in 2018 versus 23,650 ha in 2019), but they are much smaller than those for dry (pulse) pea comprising 1,385,555 ha in 2018 [27].

While ca. 80% of the dry and forage cultivars registered in Russia are of Russian origin, almost half (48%) of all vegetable cultivars originate from foreign countries [28]. It should be noted that agriculturally valuable recessive morphological mutations are more often introduced into genotypes of dry and fodder cultivars compared with vegetable ones, so Russian dry peas are more morphologically diverse.

## 3. Stem (Determinate Growth Habit)

Like many other legume crops (such as faba beans, lentil, soybean, chickpea etc.), pea plants normally grow unlimitedly (indeterminate growth pattern, IDT), producing numerous axillary inflorescences along a shoot (Figure 2A). As a result, seed maturation is prolonged over a significant period, and plants are harvested only, when pods at two or three nodes have reached the required maturity. The remaining flowers and fruits hence have no value (if not utilized as forage for animals). To rationalize plant ontogeny for practical purposes and to accelerate the maturation of pods, mutations causing determinate growth (DT) are used in breeding of different legume species [29]. Surprisingly enough, a similar morphological pattern is controlled by orthologous genes in many legumes [30]. In *P. sativum*, a recessive mutation *determinate* (*det*) causes the conversion of a shoot apical meristem into a terminal racemose inflorescence similar to the axillary one [31]. As a result, mutant plants produce two (rarely more) lateral and one terminal flower-bearing racemes and then stop growing (Figure 2B).

One of the limitations associated with the usage of mutation *det* in breeding is connected with a tight linkage between *Det* and a gene encoding a starch-branching enzyme, *R*. The first *det* mutant was induced in the 1960s from a vegetable cultivar [33] and hence had a *det r* genotype. This mutant served as the material for breeding several vegetable cultivars with the DT growth pattern, such as Pervenets, Salamat, Druzhnyi, Atlant, Kreiser, and others. These cultivars were remarkable with limited branching and more or less simultaneous seed maturation [34–36]. However, in a series of efforts to obtain *det R* recombinants in $F_2$ from crossing *det r* × *Det R*, no plants with the desired genotype were obtained, evidencing the very close localization of these genes [37]. In 1984, a spontaneous *det R* mutant was found [38], which was later used to breed several dry pea cultivars, such as Determinantnyi VSKhI. However, the *det*-mediated DT habit did not become popular enough due to a relatively low seed yield compared with more traditional IDT cultivars [37].

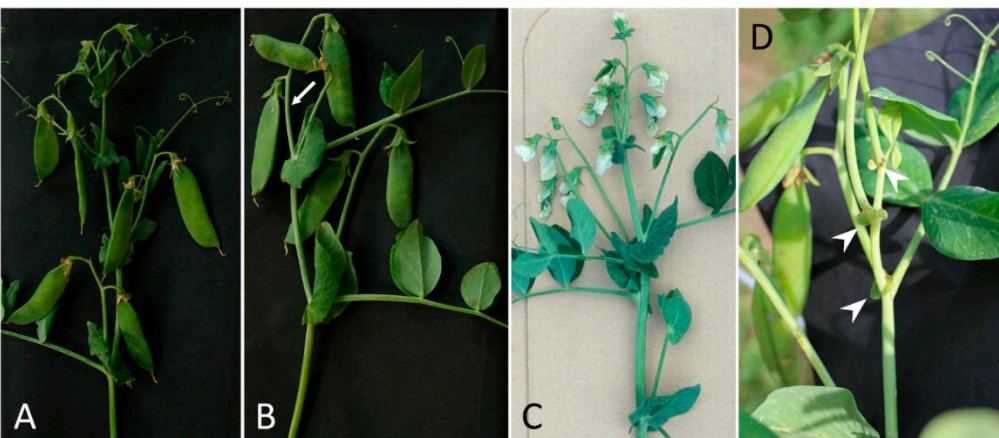

**Figure 2.** Indeterminate (**A**) and determinate growth patterns in pea conditioned by different genotypes. (**B**) *det* (arrow = terminal raceme); (**C**) 'lupinoid' (*det fa*); (**D**) *deh* (cv. Flagman; reduced stipules are indicated with arrowheads). Photos: (**A**,**B**) Daniil Shabunin; (**C**) Fedor Konovalov; (**D**) Ref. [32], reproduced with permission.

In 19th and 20th centuries, there were repeated efforts to utilize heritable forms of fasciation (recessive mutation *fa*) in pea breeding. As a result, fasciated cultivars and landraces are available in germplasm collections, such as Mummy pea from the UK or Shtambovyi 2 from the former USSR. All these forms are very susceptible to lodging due to the uneven distribution of flowers and pods along a stem and hence had no wide distribution. However, in recent years the notion of associating DT growth (*det*) and stem fasciation (*fa*) in the same genotype has been suggested. The resulting phenotype has an apical thickened flower-bearing raceme often producing more than ten flowers on short pedicels [37–39]. Such morphology resembles the many-flowered terminal racemes of *Lupinus* and is therefore called 'lupinoid' (Figure 2C). A many-flowered apical inflorescence provides the simultaneous maturation of seeds. However, 'lupinoids' are not free from lodging for the same reasons that fasciated forms are not. Some experiments have demonstrated the advantages of 'lupinoids' in conditions of superfluous humidity. These forms are within the scope of active breeding processes, often resulting in the production of recombinants with relatively improved resistance to lodging and high protein content in seeds [37–41]. However, to our knowledge, there are still no registered cultivars of the 'lupinoid' phenotype.

As an alternative to the DT associated with the mutation *det*, some Russian cultivars bear the mutation *determinate habit* (*deh*), which causes the preliminary cessation of apical meristem growth and the reduction of stipule size in the upper part of a shoot (Figure 2D). This mutation has not been characterized on a molecular level, nor even localized on linkage map. In some environments, *deh* plants produce no fewer flowering nodes than IDT forms. Moreover, the mutation *deh* is most probably semidominant rather than recessive [32]. Although the features of inheritance and pleiotropy of such DT are not yet fully characterized, numerous Russian cultivars possess *deh*-conditioned stem determinacy (Flagman, Nemchinovskii 50, Batrak).

As far as we are concerned, heritable forms of DT are not used in plant breeding beyond Russia.

## 4. Leaf

Pea plants have been serving a model for studies on the development of compound leaves (Figure 3A) for decades, for example see [42]. While several dozens of mutations may affect proper leaf ontogeny in pea [7] and bring valuable information on regulation and evolution of leaf morphogenesis, there have been repeated efforts to apply at least some of these mutations to produce cultivars with improved characteristics [42,43].

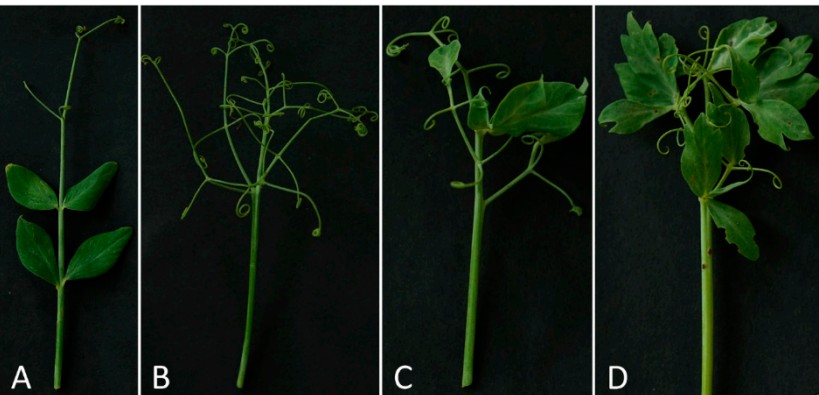

**Figure 3.** Variation of leaf morphology in different pea genotypes. (**A**) wild-type leaf (*Af Tac*^A *Uni*); (**B**) 'afila' leaf (*af Tac*^A *Uni*); (**C**) 'chameleon' leaf (*af Tac*^A *uni*^tac); (**D**) 'dissected leaflet' phenotype (*af tac*^A *Uni*). No stipules are shown. Photos: Daniil Shabunin.

The first leaf mutation with proven agricultural value was afila (*af*) promoting leaf rachis branching and preventing formation of leaflets (Figure 3B). Although a joint photosynthetic area is reduced in such leaves, *af* plants have a similar level of assimilation efficiency as *Af* forms [44]. Physiological differences between near-isogenic *af* and *Af* lines have been the subject of extensive studies. It was found that in the vegetative phase (until the production of the 11th leaf), *af* recombinants have a smaller photosynthesizing area and biomass than their *Af* analogues, but these differences become less pronounced in the course of subsequent growth [45,46]. The advantages of *af* plants can be connected to their better insolation, even in a dense stand, and their much better resistance to lodging. When highly productive cultivars are crossed with 'afila' lines, the resulting recombinants can associate a higher seed yield with a modified leaf pattern [44,47,48].

The level of photosynthetic activity can be additionally compensated by large stipules (genotype *St*). The 'afila' leaf morphology improves the resistance of plants against lodging, especially if combined with a shortened internodes (*le*) and robust stem (presumably polygenic control). As a result, since the 1970s, in both Russia and foreign countries, pea cultivars have completely changed their habit. While the old landraces and cultivars possessed a long IDT stem and wild-type leaves, more recent breeding has produced numerous cultivars with dwarf stems and the 'afila' leaf type. The principal outcome of such forms are seeds in contrast to the large amounts of straw produced by old cultivars, i.e., an improvement has occurred through the increase of the harvesting index (the ratio of grain to total shoot dry matter) [49].

The introduction of the 'afila' leaf type and dwarfism into new cultivars revolutionized the breeding of pea in Russia. As indicated in the late 1980s, the gross biomass of pea plants was almost the same as in the 1920s, but the ratio of seeds to vegetative organs increased significantly [49]. It would appear that this trend was not unique to breeding practices in the former USSR and it was probably the same worldwide. To date, in addition to numerous dry cultivars, the fodder 'afila' cultivar Alla has been bred with several vegetable cultivars, such as Parus, Darunok, Triumph, and Lider.

However, in later years there were (and still are) additional pea leaf modification programs using more morphological mutations. Since 1989, double mutants *af uni*^tac have been included in the scope of such programs. This unusual 'heterophyllous' form is being studied in the Federal Scientific Center of Legumes and Groat Crops (Oryol city), and was first found among the F$_2$ of a cross between 'tendrilled acacia' mutant (India) and cultivar Filby (Great Britain) [50] (Figure 3C).

This form was called 'chameleon', as its juvenile leaves consist mostly of tendrils, while the upper leaves have highly ramified leaf rachis and both tendrils and modified leaflets (Figure 3C). Although such a phenotype has been known about for a long time [51], it was not previously considered potentially valuable. However, it combines both an

increased number of tendrils (compared with a wild-type) and a larger photosynthesizing area than *af* plants. There evidence that *af uni*tac plants are not worse, or are even better, than more conventional *af Uni* and *Af Uni* forms [52–54]. A set of 'chameleon' lines were produced atthe Federal Scientific Center of Legumes and Groat Crops, all designated 'Az' after the name of the breeder, Anatoly Zelenov. Studies on these lines indicated that they often had a higher protein content in seeds (25–27%) than traditional cultivars. Several 'chameleon' cultivars are already registered in Russia, such as Spartak and Jaguar in Oryol [51,54], as well as Sibirskii 1, which is produced by the Institute of Cytology and Genetics, Siberian Branch of the Russ. Acad. Sci., together with the Institute of Agricul-ture of Northern Transurals.

A novel recessive mutation *tac*A, which affects leaf development, was recently de-scribed [42,55]. *tac*A mutants have a phenotype strongly resembling those of *uni*tac ('ten-drilled acacia') but these mutations were reported as non-allelic [55]. Double mutants *af tac*A produce leaves with numerous tendrils and large dissected leaflet-like laminae (Figure 3D). This 'dissected leaflet' morphotype was reported to exceed other leaf mutants in photo-synthetic parameters and produced biomass. That is why this form is discussed as highly promising for breeding new cultivars with increased resistance to lodging and improved photosynthesis characteristics [41,54–56]. There are seemingly no registered cultivars with a such phenotype to date. Gene *TAC*A still awaits its localization and identification on a molecular level.

Double mutants *af tl* have no tendrils and produce numerous small leaflet-like laminae instead ('pleiofila') [51]. Sowing a mixture of two lines or cultivars, one with 'pleiofila' and the other with 'afila' leaves, was proposed [57]. As a result, one component would produce more seeds due to improved photosynthetic features, while the other would prevent the whole system from lodging.

## 5. Nodulation and Symbiotic Nitrogen Fixation

The symbiotic relationship between pea and nitrogen-fixing bacteria is known to have a positive impact on both agriculture and the environment. Both spontaneous and induced symbiotic mutants can significantly improve the existing gene pool of pea increasing the efficiency of symbiosis [58].

Studies on the symbiotic interactions between pea and soil microflora were, for a long time, restricted to nitrogen fixing symbiosis (NFS). The genetic component of variation of this feature was first investigated in the USSR by Leonid Govorov [59] and Zinaida Razumovskaya [60]. In subsequent years, numerous mutants were induced with heritable modifications in nodulation (production of root nodules) and the development of the symbiotic system [61]. Since the 1980s, more than 40 *Sym* genes have been described, together with several *Nod* genes and some others; many of these genes have been localized on a linkage map (for review see [62]). These genes are involved in the recognition of rhizobial nodulation signals, early symbiotic signaling cascades, infection and nodulation processes, and the regulation of nitrogen fixation. Both induced mutations and naturally occurring alleles of these genes cause defects in the development of nodules and/or growth of plants in the absence of soil nitrogen. Some of the mutants defective in nitrogen fixing symbiosis are also characterized by other structural abnormalities, such as fasciation (*sym28*, *nod4*) or floral and leaf malformations (*coch*).

With the genomic revolution of the past 20 years, it has become clear that thousands of genes are expressed at relatively high levels in legume nodules [63,64]. The reverse genetics starting its path from the gene of interest and finishing with a phenotype, as distinct from classical/forward genetics, has assisted in the identification of genes with both major and minor influence on symbiotic nitrogen fixation. Operating together, forward and reverse genetics led to a breakthrough in the understanding of molecular and cellular processes underlying nitrogen fixation through the identification of ca. 200 genes involved in efficient symbiosis [65].

As a nitrogen fixed by symbiotic rhizobia contributes to the increase of plant biomass and its enrichment with nitrogen, it is quite reasonable to perform breeding for the improvement of the features of symbiotic interactions. After arbuscular mycorrhiza (AM) was discovered and found to have numerous common regulatory patterns with NFS, the tripartite symbiotic system came into focus, i.e., legume plants + arbuscular mycorrhizal fungi + nodule bacteria [66–68]. Both symbionts have a synergetic effect on the development and productivity of plants. Investigations into the practical value of this symbiosis have been led by the All-Russia Research Institute for Agricultural Microbiology (Saint Petersburg). When pea accessions capable to the maximum efficiency of symbiosis were inoculated with a mixture containing both rhizobia and mycorrhizal fungi (original breeding protocol, reviewed in [69]), the result was equivalent to the application of a full set of mineral fertilizers [70,71]. That is why studies focused on the design of highly productive and symbiotically active pea accessions were targeted at triple symbiosis. Moreover, plant welfare can be improved by the plant growth-promoting bacteria (PGPB) [72]. The effectiveness of plant interactions with beneficial soil microbes (EIBSM) has been considered as an additional agriculturally valuable trait of pea cultivars.

The production of effective plant-microbe systems through the breeding of new pea cultivars with high symbiotic potential has become a pioneering initiative in Russia, aiming at the widening of the adaptive potential of plants and the acquisition of new metabolic functions [71,73]. The novel cultivar Triumph, with an increased EIBSM, was bred for the first time in history at the Federal Scientific Center of Legumes and Groat Crops in collaboration with the All-Russia Research Institute for Agricultural Microbiology (Saint Petersburg) [74,75]. It was obtained through five backcrosses with the subsequent screening of $F_4$ hybrids for higher productivity and increased ability with respect to symbiotic interactions. Cultivar Classic (Denmark) and the accession k-8274 served as parental forms, the latter being a donor of symbiosis efficiency [76]. The EIBSM of cv. Triumph was higher than that of both parental genotypes. This cultivar has an 'afila' leaf type (Figure 4) and is characterized by stable productivity under different climate conditions, as well as by comparatively high resistance to root rots and pea weevil (*Bruchus pisorum*).

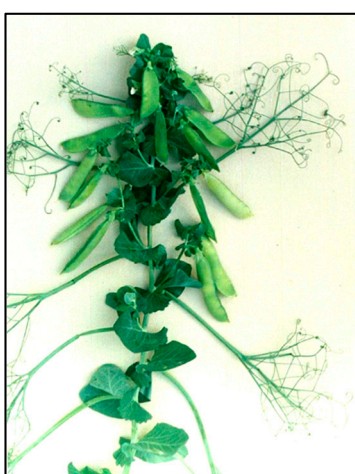

**Figure 4.** *Pisum sativum* cv. Triumph. Photo: Alexey Borisov.

The Triumph cultivar was bred exclusively by using traditional means, i.e., by using directed crosses and the phenotyping of hybrid progenies. Nowadays, whole-genome sequences of hundreds or even thousands of accessions can be obtained and used for the precise identification of genes and alleles contributing to the efficiency of symbiosis (e.g., by genome-wide association studies). Such possibilities may significantly force the further dissection of the genetic control of symbiosis and promote the breeding of peas with higher EIBSM and other legumes through MAS and other approaches [74,75].

Novel forms of pea with a higher biomass of root and symbiotic nodules can be bred through the involvement of symbiotic mutations causing super- and hypernodulation. Supernodulating mutants (e.g., *nod4*) produce a higher number of tiny nodules and are capable of active nitrogen fixation (Figure 5A). Hypernodulating mutants (such as *Nod5*) develop numerous large nitrogen-fixing nodules (Figure 5B). Recombinants bearing both mutations can be used for the further breeding of new cultivars with increased symbiosis efficiency. Planting such forms improves the fertility of the soil and can reduce the usage of mineral nitrogen fertilizers. These varieties combine relatively high productivity and the ability to enrich soil with nitrogen, so they may serve as fine precursors for planting other crop species [77].

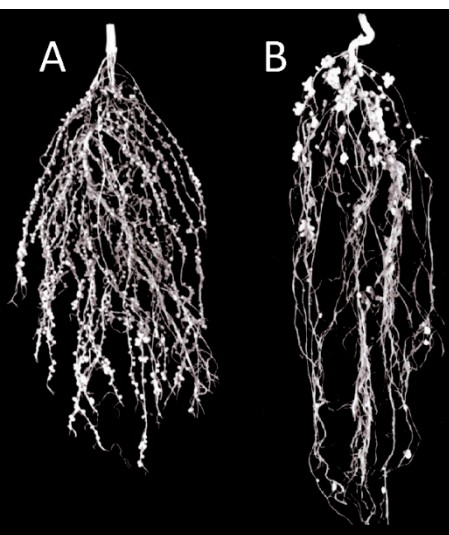

**Figure 5.** Root systems of hypernodulating (**A**) and supernodulating (**B**) mutants of pea (Ref. [58], reproduced with permission).

## 6. Collections of Germplasm with Identified Genotypes

As indicated before, morphological mutations may be a valuable source for breeding. That is why it is important to thoroughly investigate the phenotypes of mutants in comparison with their wild-types, especially in diverse environments. Germplasm collections have a special value, when accessions are characterized with respect to their genotypes, at least for genes controlling the most significant traits. Such collections serve as a source of material for breeding and different investigations. The identification of mutation on a molecular level traditionally suggests its localization on a linkage map through crosses with marker lines. Contemporary approaches (such as massive parallel sequencing) also involve the comparison between an anomalous (mutant) form with the initial one (e.g., the cultivar which the mutation was induced from) [78].

The world largest and best-known collection of pea mutants is held in the John Innes Centre (Norwich, UK). It was founded in 1994 on the basis of the collections of Swedish geneticists Herbert Lamprecht and Stig Blixt. At the moment, this collection contains ca. 3500 accessions, such as cultivars, mutants, mapping populations, and near-isogenic lines. An online web searchable catalogue of the gene list with descriptions, images, reference germplasm, and a bibliography is available [7,79].

In Russia, one of the most precisely identified and documented germplasm collections of pea belongs to the Department of Genetics at the Lomonosov Moscow State University. This collection was initiated by Sergey Gostimsky in the 1960s [80] and now includes more than 150 accessions. In addition to Russian and foreign cultivars and marker lines from different world collections, this facility includes a series of induced mutants with distortions in photosynthesis and different morphological anomalies. In addition to well-known leaf mutations (*af*, *tl*, *uni*$^{tac}$, *st* etc.), there are lines bearing novel alleles and even mutations of

yet uncharacterized genes. There are also mutants with impaired ontogeny of the stem apical meristem and inflorescence, as well as floral malformations [80].

The Vavilov Institute of Plant Genetic Resources (VIR) in Russia holds the largest European germplasm collection of pea. This collection contains more than 8000 specimens originating from 93 countries. This content is structured according to botanical and agroecological classification, breeding status (such as cultivars, local landraces, or wild-growing germplasm), as well as usage (fodder, dry etc.). Most of the accessions are characterized with respect to agriculturally valuable features, such as phenology, productivity, seed quality, and resistance to pathogens, pests and abiotic stresses. Samples from the VIR collection are in high demand as initial material for breeding and fundamental research in different biological projects [81].

In the 1980s, a special program was initiated at the VIR to identify genes controlling certain features in the germplasm collection. This collection was (and still is) actively receiving specimens with identified mutations obtained from breeders and different Russian and foreign institutions, including the John Innes Centre, as well as personally from Stig Blixt [82,83].

Most of the identified mutations affect visually recognizable features of plant morphology, such as those causing the development of 'afila' (*af*), 'acacia' (*tl*), and 'pleiofila' (*af tl*) leaf types, DT growth (*det*, *deh*), stem fasciation (*fa*), as well as non-abscising seeds (*def*). The collection of the VIR includes some accessions which bear several (six or even more) recessive mutations [84].

The representative collections of symbiotic mutants of pea are deposited in the Institute of Cytology and Genetics (Novosibirsk) [60] and in the All-Russia Research Institute for Agricultural Microbiology (Saint Petersburg), the latter being the largest depository of such germplasm in Russia. These collections include both induced mutants and forms found among natural variants with heritable alterations of different stages of nodulation.

## 7. Conclusions

In Russia, the improvement of pea for higher yields is being realized preferentially by means of traditional breeding (i.e., crosses with the subsequent selection of valuable forms) and actively involves different mutations. Many of the newly-bred cultivars' genotypes include not only well known morphological mutations (such as *a*, *af*, *le*, *r*) but also newly discovered ones. It can be stated that many of the novel mutations are readily evaluated for their breeding value. As a result, some Russian pea cultivars, especially those bred since the late 1990s, are homozygous for recessive alleles which are not (yet) utilized in pea breeding beyond Russia (*def*, *deh*, *uni*tac) and generally may have more recessive morphological mutations than foreign cultivars. For example, cv. Batrak has white flowers (*a*), shortened internodes (*le*), non-abscising seeds (*def*), the 'afila' leaf type (*af*), and determinate growth pattern (*deh*). These, along with other morphological mutations, are promising for increasing seed yield, reducing losses, and promoting the faster and more simultaneous flowering and maturation of seeds. Combinations of some mutations may be additionally beneficial. For example, double mutants *af tac*A are expected to have increased resistance to lodging and improved photosynthesis characteristics, while 'lupinoid' forms (*det fa*) seem promising in terms of increased seed yield as a result of synchronized flowering.

To date, not all agriculturally valuable mutations have been identified on a molecular level, so the MAS strategy cannot be widely applied. However, features such as anthocyanin pigmentation, leaf morphology, or internode length are easily recognizable, so the breeding process can still proceed in traditional manner, via visual phenotyping. However, the identification of respective genes and alleles is necessary for the selection of quantitative traits or symbiotic properties.

Among Russian cultivars, dry peas are most phenotypically diverse, as morphological mutations are often incorporated in their genotypes. Vegetable cultivars usually bear much fewer morphological mutations. Fodder forms predominantly have phenotypes close to wild-type, i.e., indeterminate stems with long internodes, pigmented flowers and seeds etc.

Theoretically, the possibility of combining different mutations within a single genotype has no limit. In terms of current-day pea breeding, this approach can still bear fruits, especially if it is assisted by the MAS strategy, which can make the breeding process much faster. It would appear that the biological limits of a given species, *Pisum sativum*, have not yet been fully reached. However, the unreducible problem of food security probably requires more vigorous measures, such as site-directed mutagenesis or genome editing.

**Author Contributions:** Conceptualization, A.S. and M.V.; Writing—Original Draft Preparation, A.S., M.V., E.S.; Writing—Review and Editing, A.S. and M.V.; Final formatting and submission, E.S.; Visualization, A.S. All authors have read and agreed to the published version of the manuscript.

**Funding:** The contribution of A.S. was performed as part of the state assignment № 121032500086-0 ('The study of the genetic organization of the plant genome'). The contributions of E.S. and M.V. were supported by the Budgetary Project No. 0481-2022-0002 (VIR).

**Institutional Review Board Statement:** Not applicable.

**Informed Consent Statement:** Not applicable.

**Data Availability Statement:** Not applicable.

**Acknowledgments:** The authors express their gratitude to Daniil Shabunin and Fedor Konovalov for their fine photographs of pea plants, as well as to the editorial offices of journals for permission to reproduce previously published figures.

**Conflicts of Interest:** The authors declare no conflict of interest.

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
