# Peer review of "Usage of Morphological Mutations for Improvement of a Garden Pea (Pisum sativum): The Experience of Breeding in Russia"

_agronomy, doi:10.3390/agronomy12030544_

Round 1
Reviewer 1 Report
The work is very interesting but in my opinion the classification needs to be reviewed as I indicated. The garden pea and the fodder pea are two different varieties of the genus Pisum (see references by M. Ambrose (John Innes Centre) and N. Maxted). A correct classification is especially required in a revieuw
Garden pea (P. sativum L. subsp. sativum var. sativum)
Field pea (Pisum sativum ssp.sativum var.arvense)
Abstract s
Line 11: Do the authors refer to garden peas? (P. sativum subsp. sativum var. sativum)
Introduction
Line 102: See references ... M. Ambrose and N. Maxted....Garden peas (P. sativum var. sativum) field peas (P. sativum var. arvense)
Line 104: The authors talk about fodder pea varieties, what do they mean by fodder varieties?
The genus Pisum L. consists of the following species, subspecies and varieties: P. sativum subsp. sativum var. sativum, P. sativum L. subsp. sativum var. arvense....etc...
In the introduction, the authors describe the genus Pisum L. with the species, subspecies and varieties that constitute it.
Seed and pod
Line 150: What is meant by forage varieties?
Author Response
Dear colleague,
Many thanks for your work on reviewing our manuscript and positive decision about it.
We agree that a proper taxonomic placement is important to discuss many topics. In this particular case, the situation is quite complicated. In accordance with your suggestion, we added a citation of Maxted & Ambrose (lines 106-116 in new version) and some discussion on taxonomic status of cultivars.
However, both P. sativum subsp. sativum s.l. and P. sativum subsp. sativum var. sativum can be called 'garden', as they have identical species/subspecies/variety epithet (sativum). We removed a word 'garden' from most of places in the text, where both varieties are considered (i.e. where we discuss 'cultivated peas' as a whole). However, at the moment we retained epithet 'garden' in a manuscript's title (mostly for aesthetic reasons). If you consider this improper, we are ready change a title as well.
We also made a rewording to avoid using 'variety' as a synonym of 'cultivar'. We agree it is improper.
All list of references and citations were adjusted accordingly, as two new references were introduced. The resulting manuscript is attached.
We hope that in a revised form our paper can be considered acceptable for publication. If not, we are ready to make additional corrections.
Many thanks and best regards,
Authors

Reviewer 2 Report
Very nice approach of the topic and complete description in pea from history to date. Good relevance for breeding purpose and long term strategy. Key role for phenotyping in relation with mutations.
Author Response
Dear colleague,
Many thanks for your work on reviewing our manuscript and positive decision about it. We are very glad that you found our paper interesting and acceptable for publication.
Several small changes were introduced according to the other reviewer's suggestions. For your interest, you may find an updated manuscript attached.
Many thanks and best regards,
Authors

Reviewer 3 Report
The title is clearly written. The abstract is well written. The general idea of research is well expressed. There is future scope.The introduction is sufficient to answer the scientific question. Results are well better exposed and explained. There is accurate explanation of the entire reading.The conclusions are well set. A total of 82 references, well updated.My recommendation is that the authors pay attention to the following paragraphs - Lines 42, 64, 91, 101, 250, 296, 355, 403 and 413. It is obligatory to mention the respective literary source.
Author Response
Dear colleague,
We would like to express our gratitude for your work on reviewing our paper.
We have considered most of your suggestions, as it is difficult to argue that key points need to be supported by references. Attached you may find a revised version of our manuscript. We believe that this became better due to your helpful comments and suggestions.
As for line 91 (now 92), we did not add any references confirming this thesis, as there are hardly any papers comparing a number of mutations used in breeding of different crops. However, we attempted to make this statement a bit more accurate, and now it is considering leguminous crops (previously it was about crops without getting in details). Probably in ancient crops such as maize or wheat there are even more morphological mutations involved in breeding than in pea. However, we're absolutely sure that there are no other legumes in which morphological mutations are applied for breeding as widely as in pea.
It was somewhat challenging for us to understand what statement needed to be supported with reference(s) in lines 101-102. It is about dozens of Russian papers cited below, in the main part of our review, so it is impossible to cite them in lines 101-102.
We hope that in its updated form our paper can be accepted for publication in Agronomy. Many thanks to you for your helpful suggestion.
Best regards,
Authors
